# Unsupervised Motion Representation Learning with Capsule Autoencoders

**Ziwei Xu**[†]**, Xudong Shen**[‡]**, Yongkang Wong**[†]**, Mohan S Kankanhalli**[†]

† School of Computing, National University of Singapore
‡ NUS Graduate School, National University of Singapore

{ziwei-xu, mohan}@comp.nus.edu.sg
xudong.shen@u.nus.edu, yongkang.wong@nus.edu.sg

## Abstract

We propose the Motion Capsule Autoencoder (MCAE), which addresses a key challenge in the unsupervised learning of motion representations: transformation invariance. MCAE models motion in a two-level hierarchy. In the lower level, a spatio-temporal motion signal is divided into short, local, and semantic-agnostic *snippets*. In the higher level, the snippets are aggregated to form full-length semantic-aware *segments*. For both levels, we represent motion with a set of learned transformation invariant templates and the corresponding geometric transformations by using capsule autoencoders of a novel design. This leads to a robust and efficient encoding of viewpoint changes. MCAE is evaluated on a novel Trajectory20 motion dataset and various real-world skeleton-based human action datasets. Notably, it achieves better results than baselines on Trajectory20 with considerably fewer parameters and state-of-the-art performance on the unsupervised skeleton-based action recognition task.

## 1 Introduction

Real-world movements contain a plethora of information beyond the literal sense of moving. For example, honeybees "dance" to communicate the location of a foraging site and human gait alone can reveal activities and identities [10]. Understanding these movements is vital for an artificial intelligent agent to comprehend and interact with the ever-changing world. Studies on social behavior analysis [8, 9], action recognition [59, 64], and video summarizing [60] have also acknowledged the importance of movement.

A key step towards understanding movements is to analyze their patterns. However, learning motion pattern representations is non-trivial due to (1) the curse of dimensionality from input data, (2) difficulties in modeling long-term dependencies in motion sequences, (3) high intra-class variation as a result of subject or viewpoint change, and (4) insufficient data annotation. The first two challenges have been ameliorated by the advances in keypoint detection [57], spatial-temporal feature extractors [38, 43, 50], and hierarchical temporal models [13, 49, 56]. The third and the fourth nonetheless remain hurdles and call for unsupervised transformation-invariant motion models.

Inspired by the viewpoint-invariant capsule-based representation for images [11, 17], we exploit capsule networks and introduce the Motion Capsule Autoencoder (MCAE), an unsupervised capsule framework that learns the transformation-invariant motion representation for keypoints. MCAE models motion signals in a two-level snippet-segment hierarchy. A snippet is a movement of a narrow time span, while a segment consists of multiple temporally-ordered snippets, representing a longer-time motion. In both the lower and the higher levels, the snippet capsules (SniCap) and the segment capsules (SegCap) maintain a set of templates as their identities—snippet templates and

segment templates, respectively—and transform them to reconstruct the input motion signal. While the snippet templates are explicitly modeled as motion sequences, the SegCaps are built upon the SniCaps and parameterize the segment templates *in terms of the snippet templates*, resulting in fewer parameters compared with single-layer modeling. The SniCaps and SegCaps learn transformation-invariant motion representation in their own time spans. The activations of the SegCaps serve as a high-level abstraction of the input motion signal.

The contributions of this work are as follows:

- We propose MCAE, an unsupervised capsule framework that learns a transformation-invariant, discriminative, and compact representation of motion signals. Two motion capsules are designed to generate representation at different abstraction levels. The lower-level representation captures the local short-time movements, which are then aggregated into higher-level representation that is discriminative for motion of wider time spans.

- We propose Trajectory20, a novel and challenging synthetic dataset with a wide class of motion patterns and controllable intra-class variations.

- Extensive experiments on both Trajectory20 and real-world skeleton human action datasets show the efficacy of MCAE. In addition, we perform ablation studies to examine the effect of different regularizers and some key hyperparameters of the proposed MCAE.

## 2   Related Works

**Motion Representation**   A variety of methods have been proposed to learn (mostly human) motion representation from video frames [3, 23, 28, 53], depth maps [14, 22, 27, 37, 47], key-points/skeletons [4, 21, 24, 26, 29, 33, 41, 52, 55, 58], or point clouds [6, 7]. Earlier works use handcrafted features like Fourier coefficients [47], dense trajectory features [46, 30], and Lie group representations [44]. Some works use canonical human pose [32] or view-invariant short tracklets to learn robust feature for recognition [19]. The development of deep learning brings the usage of convolution networks (ConvNet) and recurrent networks for motion representation. Simonyan *et al.* [39] proposes a two-stream ConvNet which combines video frame with optical flow. C3D [43] proposes to use 3D convolution on the spatial-temporal cubes. Srivastava *et al.* [40] uses an Long Short-Term Memory (LSTM)-based encoder to map input frames to a fixed-length vector and apply task-dependent decoders for applications such as frame reconstruction and frame prediction. The combined use of convolution module and LSTM has also been proved effective in [3, 38, 51].

A series of works [15, 18, 20, 45, 48] have been proposed to address the problem of learning viewpoint-invariant motion representation from videos or keypoint sequences. MST-AOG [48] uses an AND-OR graph structure to separate appearance of mined parts from their geometry information. Li *et al.* [20] learn view-invariant representation by extrapolating cross-view motions. View-LSTM [18] defines a view decomposition, where the view-invariant component is learned by a Siamese architecture. While most of these works exploits multi-modal input of RGB frames, depth maps or keypoint trajectories, MCAE focuses on the pure keypoint motion.

**Capsule Network**   MCAE is closely related to the Capsule Network [11], which is designed to represent objects in images using automatically discovered constituent parts and their poses. A capsule typically consists of a part identity, a set of transformation parameters (i.e. pose), and an activation. The explicit modeling of poses helps learning viewpoint-invariant part features that are more compact, flexible, and discriminative than traditional ConvNets. Capsules can be obtained via agreement-based routing mechanisms [12, 34]. More recently, Kosiorek *et al.* [17] proposed the unsupervised stacked capsule autoencoder (SCAE), which uses feed-forward encoders and decoders to learn capsule representations for images.

Apart from images, capsule network has been studied in other vision tasks. In [61, 62], capsule network is used for point cloud processing for 3D object classification and reconstruction. VideoCapsuleNet [5] proposes to generalize capsule networks from 2D to 3D for action detection in videos. Yu *et al.* [54] proposed a limited study on supervised skeleton-based action recognition using Capsule Network. Sankisa *et al.* [36] proposed to use Capsule Network for error concealment in videos.

Despite the success of capsule networks in various vision tasks, the study of capsule networks on motion representation is scarce. In this work, MCAE performs unsupervised learning of motion rep-

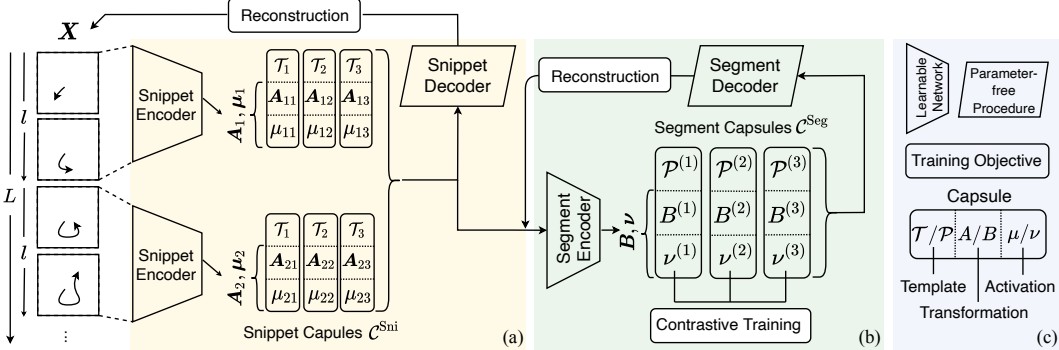

Figure 1: Overview of MCAE (best viewed in color). (a) The Snippet Autoencoder, which learns the semantic-agnostic short-time representation (snippet capsules) by reconstructing the input signal $\boldsymbol{X}$. (b) The Segment Autoencoder, which learns the semantic-aware long-time representation (segment capsules) by aggregating and reconstructing snippet capsule parameters. The activation values in segment capsules are used as semantic information for self-supervised contrastive training. (c) Meanings for different shapes and variables.

resented as coordinates rather than pixels. It aims at learning an appearance-agnostic transformation-invariant motion representation. We believe that introducing motion to the capsule network, or the other way round, provides (1) A new, robust, and efficient view into motion signals in any dimension space under the transformation-invariance assumption (while the motion and transformation in dimension space could have semantics different from their 2D/3D counterparts), and (2) proof that disentangling identity from transformation variance works not only for vision problems but a possibly larger family of time series analysis problems.

## 3 Methodology

We consider a single point[1] in $d$-dimension space. The motion of the point, i.e. a trajectory, is described by $\boldsymbol{X} = \{\boldsymbol{x}_i | i = 1, \ldots, L\}$, where $\boldsymbol{x}_i \in \mathbb{R}^d$ is the coordinates at time $i$ in a $d$-dimension space. Semantically, $\boldsymbol{X}$ belongs to a motion pattern, subject to an arbitrary and unknown geometric transformation. Given sufficient samples of $\boldsymbol{X}$, we aim to learn a discriminative (in particular, transformation-invariant) representation for those motion samples without supervision.

### 3.1 Framework Overview

We solve this problem in two steps, namely *snippet learning* and *segment learning*. Snippets and segments correspond to the lower and higher levels of how MCAE views the motion signal. Both snippets and segments are temporally consecutive subsets of $\boldsymbol{X}$, but snippets have a shorter time span than segments. In the snippet learning step, the input $\boldsymbol{X}$ is first divided into $S = L/l$ temporally non-overlapping snippets, where $l$ is the length of snippets. Each of these snippets will be mapped into a semantic-agnostic representation by the **Snippet Autoencoder**. In the segment learning step, the snippet representations are combined and fed into the **Segment Autoencoder**, where the full motion is represented as a weighted mixture of the transformed canonical representations. The segment activations are used as the motion representation for downstream tasks. An overview of the framework is shown in Fig. 1. In the following section, we delineate the details for each module and explain the training procedure.

### 3.2 Snippet Autoencoder

To encode the snippets' motion variation, we propose the Snippet Capsule (SniCap), which we denote as $\mathcal{C}^{\texttt{Sni}}$. SniCap is parameterized as $\boldsymbol{C}^{\texttt{Sni}} = \{\mathcal{T}, \boldsymbol{A}, \mu\}$, where $\mathcal{T}$, $\boldsymbol{A}$, and $\mu$ are the **snippet template**, **snippet transformation parameter**, and **snippet activation**, respectively. The snippet

---

[1]We show a way to generalize MCAE to multi-point systems in Section 4.2

template $\mathcal{T} = \{\boldsymbol{t}_i | \boldsymbol{t}_i \in \mathbb{R}^d, i = 1, ..., l\}$ describes a motion template of length $l$ and is the identity information of a SniCap. $\boldsymbol{A}$ and $\mu$ depend on the input snippet. The transformation parameter $\boldsymbol{A} \in \mathbb{R}^{(d+1)\times(d+1)}$ descries the geometric relation between the input snippet and the snippet template. The snippet activation $\mu \in [0,1]$ denotes whether the snippet template is activated to represent the input snippet.

**Snippet Encoding/Decoding**  For a given snippet $\boldsymbol{x}_{i:i+l}$, the snippet module performs the following steps: (1) encode motion properties with Snippet Encoder into SniCaps, and (2) decode SniCaps to reconstruct the original $\boldsymbol{x}_{i:i+l}$. For the encoding step, a 1D-ConvNet $f_{\text{CONV}}$ is used to extract the motion information from $\boldsymbol{x}_{i:i+l}$ and predict SniCap parameters, i.e. $\{(\boldsymbol{A}_j, \mu_j)|j = 1, \ldots, N\} = f_{\text{CONV}}(\boldsymbol{x}_{i:i+l})$ where $N$ is the number of SniCaps. The range of $\mu$ is confined by a sigmoid activation function. For the decoding step, we first apply the transformation $\boldsymbol{A}$ to the snippet templates as

$$\begin{pmatrix} \hat{\boldsymbol{t}}_{ij} \\ 1 \end{pmatrix} = \boldsymbol{A}_i \begin{pmatrix} \boldsymbol{t}_j \\ 1 \end{pmatrix}, \quad i = 1, \ldots, N, \quad j = 1, \ldots, l. \tag{1}$$

Then, the transformed templates from different SniCaps are mixed, according to their activations, and the corresponding reconstructed input is

$$\hat{\boldsymbol{x}}_j = \sum_{i=1}^{N} \mu_i \hat{\boldsymbol{t}}_{ij}, \quad j = 1, \ldots, l, \tag{2}$$

where $\hat{\boldsymbol{t}}_{ij}$ indicates the transformed coordinate of the $i^{th}$ SniCap at $j^{th}$ time step.

### 3.3   Segment Autoencoder

The motion information encoded in SniCaps is agnostic to the segment level motion patterns. This makes it less biased towards the training data domain. However, its utility on high-level applications, such as activity analysis or motion classification, is greatly undermined. For example, consider Fig. 2(a) as a reference "triangle" trajectory. Fig. 2(b) illustrates a possible intra-class variation. Since the two trajectories differ greatly in their local movement, they could be considered as different classes without transformation-invariant information from the full trajectory.

Hence, we introduce a segment encoder to gain a holistic understanding of motion and encapsulate such information in the segment capsules (SegCap). A segment is a motion of length $L$ (generally the segment length do not have to be the same as the signal length) and can be interpreted as $S = L/l$ consecutive non-overlapping snippets. A SegCap is parameterized as $\boldsymbol{C}^{\text{Seg}} = \{\mathcal{P}, \boldsymbol{B}, \nu\}$, where $\mathcal{P}$, $\boldsymbol{B}$, and $\nu$ are the **segment template**, **segment transformation parameter**, and **segment activation**, respectively.

Different from the SniCap, whose template is explicitly a motion sequence, the SegCap parameterizes the segment template $\mathcal{P}$ in terms of the $N$ snippet templates. Specifically, $\mathcal{P} = \{(\boldsymbol{P}_i, \boldsymbol{\alpha}_i) \mid i = 1, \ldots, S\}$, where $\boldsymbol{P}_i \in \mathbb{R}^{N\times(d+1)\times(d+1)}$ and $\boldsymbol{\alpha}_i \in \mathbb{R}^N$. Each $\boldsymbol{P}_{ij} \in \mathbb{R}^{(d+1)\times(d+1)}$ ($j \in [N]$ additionally

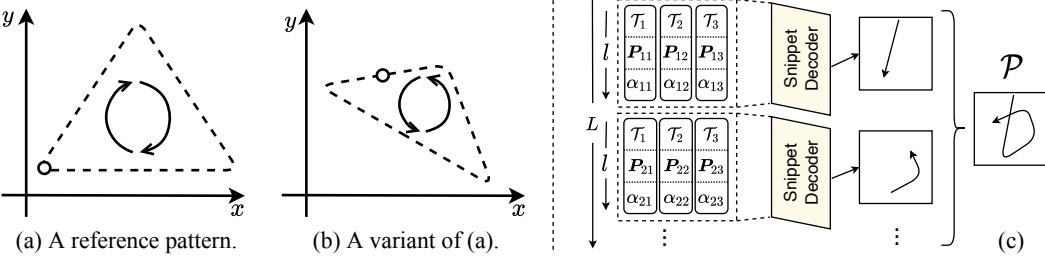

(a) A reference pattern.   (b) A variant of (a).             (c)

Figure 2: (a) and (b) show a reference motion pattern and a variant of it. The circle and the arrow shows the start and the direction of motion respectively. (c) Interpretation of a segment template $\mathcal{P}$. $\mathcal{P}$ is functionally the same as $S$ snippet parameters $(\boldsymbol{A}, \boldsymbol{\mu})$. When combined with $\mathcal{T}$, it can be decoded into an $L$-long sequence. The segment autoencoder maintains multiple segment templates, which can be transformed and mixed to reconstruct the input snippet parameters.

indexes the first dimension of $\boldsymbol{P}_i$) describes how the $j^{th}$ snippet template is aligned to form the $i^{th}$ snippet of the segment template. The weight $\alpha_{ij}$ ($j \in [N]$ additionally indexes the elements of $\boldsymbol{\alpha}_i$) controls the importance of $j^{th}$ snippet template for the $i^{th}$ snippet of the segment template. In other words, $(\boldsymbol{P}_i, \boldsymbol{\alpha}_i)$ describes how the $N$ snippet templates are used to construct an $l$-long snippet and a SegCap requires $S$ such parameters to describe an $L$-long segment template. Fig. 2(c) illustrates the interpretation of $\mathcal{P}$. $\boldsymbol{B}$ and $\nu$ are dependent on the input. $\boldsymbol{B} \in \mathbb{R}^{(d+1)\times(d+1)}$ is a transformation on $\boldsymbol{P}$, and $\nu \in [0, 1]$ is the activation of the SegCap. The segment template $\mathcal{P}$ is fixed for a SegCap w.r.t the training domain.

**Segment Encoding/Decoding**  Assume we have $M$ SegCaps with which we hope to reconstruct the low-level motion encoded in the SniCap parameters. This is equivalent to reconstructing all the data-dependent SniCap parameters $[\mathcal{C}_1^{\mathtt{Sni}}, \ldots, \mathcal{C}_S^{\mathtt{Sni}}]$, where $\mathcal{C}_i^{\mathtt{Sni}} = \{(\boldsymbol{A}_{ij}, \mu_{ij}) \mid j = 1, \ldots, N\}$ is the set of SniCap parameters for the $i^{th}$ snippet. To obtain the SegCap parameters, we first flatten each SniCap's $\boldsymbol{A}$ into a vector and concatenate it with its corresponding $\mu$. Then we encode the $S$-long sequence of flattened SniCap parameters with an LSTM model $f_{\mathtt{LSTM}}$ shared by all SegCaps, and $M$ fully-connected layers (one for each SegCap) to produce $\{\boldsymbol{B}, \nu\}$. Formally,

$$
\boldsymbol{h} = f_{\mathtt{LSTM}}\Big( \big[ \mathcal{C}_1^{\mathtt{Sni}}, \ldots, \mathcal{C}_S^{\mathtt{Sni}} \big] \Big),
$$
$$
\{ \boldsymbol{B}^{(k)}, \nu^{(k)} \} = f_{\mathtt{FC}}^{(k)}(\boldsymbol{T}, \boldsymbol{h}), \quad k = 1, \ldots, M,
$$
(3)

where $\boldsymbol{T} = \{ \mathcal{T}_i | i = 1, \ldots, N \}$, and superscript $(k)$ refers to the $k^{th}$ SegCap. The transformation and activation parameters are then applied to $\mathcal{P}$ to reconstruct snippet parameters

$$
\hat{\boldsymbol{P}}_{ij}^{(k)} = \boldsymbol{B}^{(k)} \times \boldsymbol{P}_{ij}^{(k)}, \quad i = 1, \ldots, S, \quad j = 1, \ldots, N, \quad k = 1, \ldots, M,
$$
$$
\hat{\mathcal{C}}_i^{\mathtt{Sni}} = (\hat{\boldsymbol{A}}_i, \hat{\boldsymbol{\mu}}_i) = \Big( \sum_{k=1}^{M} \nu^{(k)} \hat{\boldsymbol{P}}_i^{(k)}, \sum_{k=1}^{M} \nu^{(k)} \boldsymbol{\alpha}_i^{(k)} \Big), \quad i = 1, \ldots, S,
$$
(4)

where $\hat{\boldsymbol{A}}_i \in \mathbb{R}^{N \times (d+1) \times (d+1)}$ and $\hat{\boldsymbol{\mu}}_i \in \mathbb{R}^N$ are the reconstructed snippet transformation and activation of the snippet templates for the $i^{\text{th}}$ snippet. Note that $S = L/l$, which means $f_{\mathtt{LSTM}}$ can have a much smaller footprint than a recurrent network that handles the whole $L$-long sequence.

The above formulation enables SegCap to learn a transformation-invariant representation of motion. Intuitively, $\mathcal{P}$ describes snippet-segment relation, and $\boldsymbol{B}$ can be regarded as the spatial relation between a segment template pattern and the observed trajectory. The segment activation $\boldsymbol{\nu} \in \mathbb{R}^M$ reveals the semantics of the input trajectory and can be used for self-supervised training.

### 3.4  Training

As delineated in Section 3.2 and 3.3, SniCap and SegCap play different roles by capturing information at two different abstraction levels. SniCap focuses on short-time motion while SegCap is defined upon SniCap to model long-time semantic information. Hence, the two autoencoders are trained using different objective functions.

The only objective of the snippet autoencoder is to faithfully reconstruct the original input. Therefore, for a training sample $\boldsymbol{X} = \{ \boldsymbol{x}_i | i = 1, \ldots, L \}$, we use a self-supervised reconstruction loss:

$$
\mathcal{L}_{\mathtt{Rec}}^{\mathtt{Sni}} = \sum_{i=1}^{L} ||(\hat{\boldsymbol{x}}_i - \boldsymbol{x}_i)||_2^2,
$$
(5)

where $\hat{\boldsymbol{x}}_i$ denotes the reconstructed coordinate following Equation (2).

The segment autoencoder's primary goal is to reconstruct the input SniCap parameters, hence the reconstruction loss

$$
\mathcal{L}_{\mathtt{Rec}}^{\mathtt{Seg}} = \sum_{i=1}^{S} ||(\hat{\boldsymbol{A}}_i - \boldsymbol{A}_i)||_2^2 + ||(\hat{\boldsymbol{\mu}}_i - \boldsymbol{\mu}_i)||_2^2.
$$
(6)

Furthermore, we use unsupervised contrastive training to learn semantic meaningful activations $\boldsymbol{\nu}$. For a batch of $B$ samples, the contrastive loss is

$$
\mathcal{L}_{\mathtt{Con}}^{\mathtt{Seg}} = -\frac{1}{B} \sum_{i=1}^{B} \log \frac{\exp\big( \mathtt{cossim}(\boldsymbol{\nu}_i', \boldsymbol{\nu}_i'')/\tau \big)}{\sum_{j=1}^{B} \exp\big( \mathtt{cossim}(\boldsymbol{\nu}_i', \boldsymbol{\nu}_j'')/\tau \big)},
$$
(7)

where $\tau = 0.1$ is the temperature used for all experiments, $\boldsymbol{\nu}'_i$ and $\boldsymbol{\nu}''_i$ is the segment activation of sample $\boldsymbol{X}'_i$ and $\boldsymbol{X}''_i$, respectively. Here, $\boldsymbol{X}'_i$ and $\boldsymbol{X}''_i$ are the spatial-temporally disturbed versions of $\boldsymbol{X}_i$. The disturbance is dataset-dependent and will be discussed in the supplementary material.

In additional to the above loss terms, we impose two regularizers: a smoothness constraint on reconstructed sequence, and a sparsity regularization on the segment activations

$$\mathcal{L}_{\mathtt{Smt}}^{\mathtt{Reg}} = \sum_{i=2}^{L} ||\hat{\boldsymbol{x}}_i - \hat{\boldsymbol{x}}_{i-1}||_2^2, \quad \mathcal{L}_{\mathtt{Sps}}^{\mathtt{Reg}} = ||\boldsymbol{\nu}||_2^2. \tag{8}$$

The final training objective is:

$$\mathcal{L} = \lambda^{\mathtt{Sni}} \mathcal{L}_{\mathtt{Rec}}^{\mathtt{Sni}} + \lambda^{\mathtt{Seg}} \mathcal{L}_{\mathtt{Rec}}^{\mathtt{Seg}} + \mathcal{L}_{\mathtt{Con}}^{\mathtt{Seg}} + 0.5 \mathcal{L}_{\mathtt{Smt}}^{\mathtt{Reg}} + 0.05 \mathcal{L}_{\mathtt{Sps}}^{\mathtt{Reg}}, \tag{9}$$

where the weights are empirically determined. $\lambda^{\mathtt{Sni}}$ and $\lambda^{\mathtt{Seg}}$ are dependent on the target dataset and the remaining weights are consistent for all datasets in our experiment.

## 4 Experiments

In this section, we first assess the proposed MCAE on a synthetic motion dataset to show its ability in learning transformation-invariant robust representations. Then, we generalize MCAE to multi-point systems and show its efficacy on real-world skeleton-based human action datasets. All unsupervised accuracies are produced by an auxiliary linear classifier that is trained on the motion representation learned by MCAE or the baselines, but whose gradient is blocked from back-propagating to the model. We report the mean accuracy and standard error based on three runs with random initialization. The experiments are run on an NVIDIA Titan V GPU, where we use a batch size of 64, and the Adam [16] optimizer with a learning rate of $10^{-3}$. Please refer to the supplementary material for details.

### 4.1 Learning from Synthesized Motion

**The Trajectory20 Dataset** Although commonly used in the motion representation learning literature, datasets like movingMNIST [40] are innately *linear* and have limited motion variations. Moreover, the prediction-oriented setting makes it difficult to examine the motion category of each trajectory. In this paper, we introduce the Trajectory20 (T20), a synthetic trajectory dataset based on 20 distinct motion patterns (as shown in Fig. 3). Each sample in T20 is a 32-step-long sequence of coordinates in $[-1, 1]^2$. In the data generating process, a motion template is randomly picked, randomly rotated and scaled, and translated to a random position to produce a trajectory. A closed trajectory (marked blue in Fig. 3) starts at a random point on the trajectory and end at the same point, whereas an open trajectory (marked yellow in Fig. 3) starts at an either end's vicinity. The randomized generating process ensures the trajectories are controllably diverse in scale, rotation, and position. The training data is generated on-the-fly and a fixed test set of 10,000 samples is used for evaluation. Examples of T20 are shown in the supplementary material.

**Ablation Study** We perform an ablation study of MCAE on T20 to examine the effect of different regularizers and three key hyperparameters: snippet length $l$, the numbers of SniCap

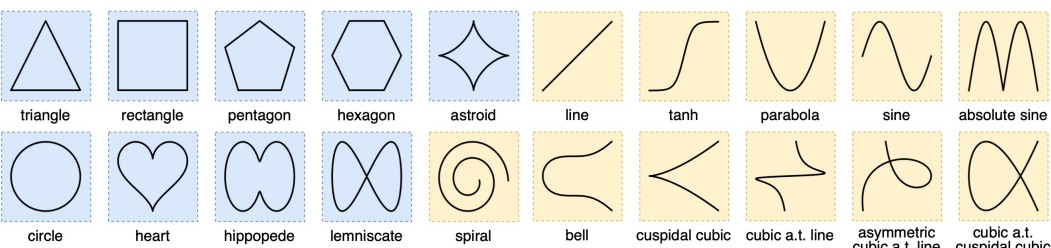

Figure 3: The 20 motion patterns in the Trajectory20 (T20) dataset. a.t. is short for "asymptotic to".

(#Sni) and SegCap (#Seg). The result is shown in Table 1. The length of snippet $l$ plays a vital role in learning a useful representation. A very small $l$ results in a narrow receptive field for snippet capsules, which makes it less useful for inferring semantics of the whole sequence. At the other end, a large $l$ makes snippets challenging to reconstruct. The numbers of SniCap and SegCap also have major effect on the outcome. Too few SniCaps makes it difficult to reconstruct the input motion signal. Too few SegCaps undermines the expressiveness of the segment autoencoder. Too many SniCaps could cause difficulty in learning proper alignments between SegCaps and SniCaps. Both degrade the quality of the learned features. Moreover, increasing #Seg from 80 to 128 does not bring further improvements. As the result shows, $(l, \text{#Sni}, \text{#Seg}) = (8, 8, 80)$ performs well and we will use it in all experiments below. As for the regularizers, while both regularizers improve the performance, the sparsity regulation ($\mathcal{L}_{\text{Sps}}^{\text{Reg}}$) on segment activation is more helpful for learning discriminative features.

Table 1: Ablation study on T20.

| Reg. | $l$ | #Sni | #Seg | Acc. (%) |
|---|---|---|---|---|
| | 8 | 8 | 80 | $69.30 \pm 0.76$ |
| | 4 | 8 | 80 | $41.01 \pm 8.81$ |
| | 16 | 8 | 80 | $45.83 \pm 8.36$ |
| Full | 8 | 2 | 80 | $64.02 \pm 2.10$ |
| | 8 | 4 | 80 | $68.17 \pm 0.36$ |
| | 8 | 16 | 80 | $48.11 \pm 1.60$ |
| | 8 | 8 | 32 | $42.36 \pm 3.15$ |
| | 8 | 8 | 64 | $63.94 \pm 1.41$ |
| | 8 | 8 | 128 | $69.44 \pm 1.69$ |
| w/o $\mathcal{L}_{\text{Smt}}^{\text{Reg}}$ | 8 | 8 | 80 | $67.60 \pm 1.69$ |
| w/o $\mathcal{L}_{\text{Sps}}^{\text{Reg}}$ | 8 | 8 | 80 | $65.92 \pm 1.63$ |

**Motion Classification**  We compare MCAE with the following baseline models, namely KMeans, DTW-KMeans, $k$-Shape [31], LSTM and 1D-Conv[2]. KMeans, DTW-KMeans, and $k$-Shape are parameter-free time series clustering algorithms. Briefly, KMeans uses Euclidean distance to measure the similarity between signals. DTW-KMeans normalizes input signals using dynamic time warping [35], and performs KMeans on the normalized signals. $k$-Shape uses cross-correlation based distance measure to cluster time series. We use the implementation by `tslearn` [42] for the three clustering methods. LSTM, 1D-Conv, and MCAE are used as backbone networks, which take the raw coordinate sequence as input and output a feature vector of a pre-defined dimension. The feature vector is used for contrastive learning following Equation (7). The corresponding accuracy reflects the quality of the learned representation.

For LSTM and 1D-Conv backbone, different numbers of hidden units/channels have been explored (shown as Hidden Param. in Table 2), which has resulted in different model sizes (measured by #Param. in Table 2).

As shown in Table 2, since the spatial variance (e.g. viewpoint changes) within motion signal cannot be directly captured by temporal warping/correlation, all the three parameter-free clustering methods perform poorly on T20. On the other hand, with considerably fewer parameters, MCAE outperforms LSTM and 1D-CNN by a large margin. This provides quantitative evidence that MCAE can capture the transformation-invariant semantic information more efficiently than the compared baselines.

Table 2: Unsupervised learning performance of MCAE and baselines on T20.

| | Hidden Param. | #Param. | Acc. (%) |
|---|---|---|---|
| KMeans | – | – | $8.57 \pm 0.04$ |
| DTW-KMeans | – | – | $9.12 \pm 0.20$ |
| $k$-Shape [31] | – | – | $12.94 \pm 0.34$ |
| | 128 | 600k | $29.17 \pm 2.45$ |
| | 256 | 669k | $40.03 \pm 0.57$ |
| LSTM | 512 | 805k | $45.59 \pm 1.37$ |
| | 1,024 | 1,078k | $53.47 \pm 1.52$ |
| | 2,048 | 1,625k | $54.32 \pm 0.55$ |
| | 128 | 588k | $44.78 \pm 0.57$ |
| 1D-Conv | 256 | 787k | $53.69 \pm 0.53$ |
| | 512 | 1,185k | $57.57 \pm 0.56$ |
| | 1,024 | 1,982k | $57.58 \pm 0.08$ |
| | (#Sni, #Seg) | #Param. | Acc. (%) |
| MCAE | (8, 80) | **277k** | $\mathbf{69.30 \pm 0.76}$ |

## 4.2 Generalizing to Multiple Points

The MCAE running on T20 dataset handles a single moving point while most real-world problems involve multiple points. This section presents MCAE-MP, a naive but effective extension of

---

[2] Architectures of LSTM and 1D-Conv are detailed in the supplementary material.

Table 3: Performance (%) for skeleton-based action classification. Column "Mod." shows the data modality, where "S" indicates skeleton and "D" indicates depth map. Column "Cls." shows the auxiliary classifier used for supervised training. We also report supervised SOTAs for completeness.

| | Model | Mod. | Cls. | NTU60 XSUB | NTU60 XVIEW | NTU120 XSUB | NTU120 XSET | NW-UCLA V1&V2 $\to$ V3 |
|---|---|---|---|---|---|---|---|---|
| Unsupervised | Luo *et al.* [27] | S+D | SLP | 61.4 | 53.2 | – | – | 50.7 |
| | Li *et al.* [20] | S+D | SLP | 68.1 | 63.9 | – | – | 62.5 |
| | SeBiReNet [29] | S | LSTM | – | 79.7 | – | – | 80.3 |
| | LongT GAN [63] | S | SLP | 39.1 | 48.1 | – | – | 74.3 |
| | MS$^2$L [24] | S | SLP | 52.6 | – | – | – | 76.8 |
| | CAE+ [33] | S | SLP | 58.5 | 64.8 | 48.6 | 49.2 | – |
| | MCAE-MP (SLP) | S | SLP | **65.6** | 74.7 | **52.8** | **54.7** | 83.6 |
| | P&C [41] | S | 1-NN | 50.7 | 76.1 | – | – | **84.9** |
| | MCAE-MP (1-NN) | S | 1-NN | 51.9 | **82.4** | 42.3 | 46.1 | 79.1 |
| Supv. | DropGraph [2] | S | – | 90.5 | 96.6 | 82.4 | 84.3 | 93.8 |
| | JOLO-GCN [1] | S | – | 93.8 | 98.1 | 87.6 | 89.7 | – |

MCAE, to enable processing motion for multi-point systems. Such motion can be described as $\mathcal{X} = \{\boldsymbol{X}_i | i = 1, \ldots, K\}$, where $K$ is the number of moving points. The extension works as follows:

1. The $K$ moving points are processed separately by an MCAE. This results in $K$ segment activation vectors $\{\boldsymbol{\nu}_i, | i = 1, \ldots, K\}$.

2. The $K$ activation vectors are concatenated into a single representation $\boldsymbol{\nu} \in \mathbb{R}^{KM}$, which is used for unsupervised learning following Equation (9).

**Skeleton-based Human Action Recognition**  We apply MCAE-MP to solve the unsupervised skeleton-based action recognition problem, where a human skeleton is a system consisting of multiple moving joints (points). Three widely-used datasets are used for evaluation: NW-UCLA [48], NTU-RGBD60 (NTU60) [37], and NTU-RGBD120 (NTU120) [25]. The three datasets consist of sequences with 1 or 2 subjects whose movement is measured in 3D space. For NW-UCLA, we follow previous works [41] to train the model on view 1 and view 2, and test the model on view 3. For NTU60, we follow the official data split for the cross-subject (XSUB) and cross-view (XVIEW) protocols. The similar is implemented on NTU120 for the cross-subject (XSUB) and cross-setting (XSET) protocol. For ease of implementation, we project the 3D sequence into three orthonormal 2D spaces and use an MCAE defined on the 2D space to process the three views of the sequences. Then the segment activations from the three views are concatenated to form the representation. Four types of disturbance are introduced for contrastive learning, namely jittering, spatial rotation, masking, and temporal smoothing. The readers are referred to the supplementary material for details.

The classification accuracy is put into three groups in Table 3. In the first group are the prior works that are not directly comparable as they use depth map [20, 27] or stronger auxiliary classifiers for supervised training [29]. In the second group, where our model is marked as MCAE-MP (SLP), a single layer perceptron (SLP) is trained as the auxiliary classifier with backbone parameters frozen. In the third group, where our model is marked as MCAE-MP (1NN), a 1-nearest-neighbor classifier is used instead of an SLP. For completeness, the fourth group shows state-of-the-art results from supervised methods. Although MCAE-MP is a naive extension as it encodes joints separately and largely ignores their interactions, it achieves better or competitive performance compared with the baselines. Notably, on NTU60-XVIEW and NTU120-XSET where the training set and test set have different viewpoints, our model outperforms baselines by a clear margin thanks to the capsule-based representation which effectively captures viewpoint changes as transformations on input.

## 4.3   What does MCAE Learn?

To better understand what is encoded, we plot the learned snippet templates $\mathcal{T}$ and segment templates $\mathcal{P}$ in Fig. 4. Note that $\mathcal{T}$ are initialized as random straight lines, and $\mathcal{P}$ are initialized as arbitrary patterns composed randomly of $\mathcal{T}$. As shown in Fig. 4a, the snippets are mainly simple lines and hook-like curves that does not carry semantic information. Segment templates in Fig. 4b, how-

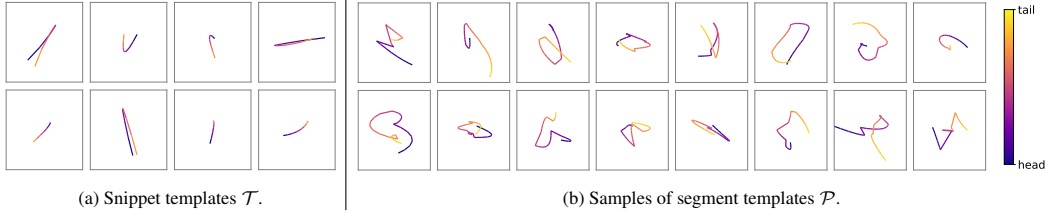

(a) Snippet templates $\mathcal{T}$.    (b) Samples of segment templates $\mathcal{P}$.

Figure 4: Templates learned from Trajectory20 dataset. Color indicates time.

ever, bear some resemblance to the patterns shown in Fig. 3. This suggests that semantic-agnostic snippets are being aggregated into semantic-aware segments.

We proceed to explore the information in SegCaps. In particular, we would like to see if SegCaps have learned transformation-invariant information. To this purpose, we randomly sample a trajectory from T20 dataset. The trajectory is first normalized so that its centroid is at $(0, 0)$, then rotated clockwise by an angle $\theta$, and finally fed into the model. We examine the segment templates with the highest activation values (which reflects the trajectory's semantics) and calculate the rotation angle $\phi$ from those templates' parameter $\boldsymbol{B}$. As shown in Table 4, the calculated $\phi$ reveals two types of segments templates as we rotate the input. One type yields constant $\phi$ (e.g. segment ID 2 for sample "absolute sine"), which indicates its rotation-invariance, the other has $\phi$ that changes monotonically with $\theta$ (e.g. segment ID 8 for sample "hexagon"), which shows its rotation-awareness. As for the activation values, samples from different categories activate different sets of segment templates. Meanwhile, the same sample under different rotation angle $\theta$ gives stable segment template activations, despite some changes which are found to have no effect on the classification result.

We do a similar study on the translation component $(x, y)$, where we translate the input by $(\Delta x, \Delta y)$. As shown in Table 5, $(x, y)$ changes monotonically with $(\Delta x, \Delta y)$ while the activated segment templates remain stable. These results prove that the semantics and transformation information has been

Table 4: Top-5 segment templates (sorted by segment activation $\nu$ then segment ID for better visualization), and the rotation $\phi$ calculated from their parameters $\boldsymbol{B}$. Bold IDs are segments repeating across different $\theta$.

| Input | $\theta = -10°$ | | $\theta = -5°$ | | $\theta = 0°$ | | $\theta = 5°$ | | $\theta = 10°$ | |
|---|---|---|---|---|---|---|---|---|---|---|
| | ID | $\phi$ | ID | $\phi$ | ID | $\phi$ | ID | $\phi$ | ID | $\phi$ |
| hexagon | **2** | 6.3 | **2** | 6.7 | **2** | 6.8 | **2** | 7.0 | **2** | 7.1 |
| | **8** | 6.9 | **8** | 9.0 | **8** | 11.2 | **8** | 13.9 | **8** | 16.5 |
| | **12** | 54.9 | **12** | 55.5 | **12** | 55.8 | **12** | 56.5 | **12** | 56.8 |
| | **37** | -20.8 | **37** | -19.8 | **37** | -18.9 | **37** | -17.9 | **37** | -16.9 |
| | **66** | 50.2 | **66** | 52.5 | **66** | 55.4 | **66** | 59.0 | **66** | 62.4 |
| abs_sine | **2** | 12.1 | **2** | 12.3 | **2** | 12.2 | **2** | 12.1 | **2** | 11.9 |
| | 7 | 8.2 | 5 | -10.7 | 5 | -10.1 | 5 | -9.9 | 7 | 17.2 |
| | 33 | 65.1 | 7 | 10.7 | 7 | 13.4 | 7 | 15.4 | 32 | -9.7 |
| | **37** | -22.9 | **37** | -22.3 | **37** | -21.8 | **37** | -21.3 | **37** | -19.9 |
| | **46** | 45.7 | **46** | 47.5 | **46** | 48.6 | **46** | 50.2 | **46** | 51.6 |

Table 5: Top-5 segment templates (sorted by segment activation $\nu$ then segment ID for better visualization), and the translation $(x, y)$ calculated from their parameters $\boldsymbol{B}$.

| Input | $(\Delta x, \Delta y) = (-0.2, 0)$ | | | $(\Delta x, \Delta y) = (-0.1, 0)$ | | | $(\Delta x, \Delta y) = (0, 0)$ | | | $(\Delta x, \Delta y) = (0, 0.1)$ | | | $(\Delta x, \Delta y) = (0, 0.2)$ | | |
|---|---|---|---|---|---|---|---|---|---|---|---|---|---|---|---|
| | ID | $x$ | $y$ | ID | $x$ | $y$ | ID | $x$ | $y$ | ID | $x$ | $y$ | ID | $x$ | $y$ |
| hexagon | **2** | 0.05 | 0.18 | **2** | 0.17 | 0.19 | **2** | 0.27 | 0.19 | **2** | 0.28 | 0.28 | **2** | 0.27 | 0.37 |
| | **8** | 0.01 | -0.07 | **8** | 0.09 | -0.06 | **8** | 0.18 | -0.04 | **8** | 0.19 | 0.04 | **8** | 0.19 | 0.12 |
| | **12** | -0.09 | 0.13 | **12** | 0.00 | 0.13 | **12** | 0.09 | 0.13 | **12** | 0.09 | 0.23 | **12** | 0.09 | 0.32 |
| | **37** | 0.10 | -0.11 | **37** | 0.18 | -0.11 | **37** | 0.27 | -0.11 | **37** | 0.27 | -0.03 | **37** | 0.27 | 0.05 |
| | **66** | -0.12 | 0.16 | **66** | -0.03 | 0.16 | **66** | 0.05 | 0.17 | **66** | 0.06 | 0.26 | **66** | 0.06 | 0.35 |
| abs_sine | **2** | 0.04 | 0.2 | **2** | 0.14 | 0.19 | **2** | 0.24 | 0.19 | **2** | 0.24 | 0.28 | **2** | 0.23 | 0.38 |
| | **5** | -0.01 | 0.30 | **5** | 0.07 | 0.29 | **5** | 0.16 | 0.29 | **5** | 0.16 | 0.38 | **5** | 0.15 | 0.46 |
| | **7** | 0.20 | -0.16 | **7** | 0.28 | -0.16 | **7** | 0.37 | -0.15 | **7** | 0.36 | -0.06 | **7** | 0.36 | 0.04 |
| | **37** | 0.04 | -0.17 | **37** | 0.12 | -0.16 | **37** | 0.21 | -0.16 | **37** | 0.20 | -0.07 | **37** | 0.20 | 0.01 |
| | **46** | 0.02 | 0.01 | **46** | 0.13 | 0.02 | **46** | 0.23 | 0.04 | **46** | 0.23 | 0.13 | **46** | 0.22 | 0.23 |

encoded separately in the segment activation $\nu$ and transformation parameters $B$. In other words, the encoded semantic information is robust against geometric transformations.

## 5    Conclusion

In this paper, we introduce MCAE, a framework that learns robust and discriminative representation for keypoint motion. To resolve the intra-class variation of motion, we propose to learn a compact and transformation-invariant motion representation using a two-level capsule-based representation hierarchy. The efficacy of the learned representation is shown through an experimental study on synthetic and real-world datasets. The output of MCAE could serve as mid-level representation in other frameworks, e.g. Graph Convolution Network, for tasks that involve more context than classification. We anticipate this work to inspire further studies that apply capsule-based models to other time series processing tasks, such as joint modeling of visual appearance and motion in video. The source code and the T20 dataset of our research are accessible at `https://github.com/ZiweiXU/CapsuleMotion`.

Motion analysis techniques are in the foreground of the misuse of machine learning methods, among which adverse societal impacts and privacy breach are two major concerns. Regarding the societal impacts, admittedly, our method has both upside and downside. On one hand, a transformation-invariant motion representation enables us better decode the information implicit in the trajectory, which has applications for example in ethology. On the other hand, it could also be misused in mass surveillance. Appropriate boundaries of use and ethical review are required to prevent potential malicious applications. Regarding the privacy concerns, our method isolates the subjects' motion from their sensitive information, such as gender and race.

## Acknowledgments and Disclosure of Funding

This research/project is supported by the National Research Foundation, Singapore under its Strategic Capability Research Centres Funding Initiative. Any opinions, findings and conclusions or recommendations expressed in this material are those of the author(s) and do not reflect the views of National Research Foundation, Singapore. The computational work for this article was partially performed on resources of the National Supercomputing Centre, Singapore (`https://www.nscc.sg`).

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
