# Appendix for Unsupervised Motion Representation Learning with Capsule Autoencoders

**Ziwei Xu[†], Xudong Shen[‡], Yongkang Wong[†], Mohan S Kankanhalli[†]**

† School of Computing, National University of Singapore
‡ NUS Graduate School, National University of Singapore

{ziwei-xu, mohan}@comp.nus.edu.sg
xudong.shen@u.nus.edu, yongkang.wong@nus.edu.sg

## A  Table of Notations

We show in the table below the notations grouped by the modules. The values used in our implementation are shown if applicable.

Table A1: Table of Notations

| Model | |
|---|---|
| $\boldsymbol{X}$ | Input motion signal |
| $\boldsymbol{x}_t$ | The $t^{th}$ time step of the input signal |
| **Sizes** | |
| $L = 32$ | Input length |
| $l = 8$ | Snippet length |
| $S$ | Number of snippets, $S = L/l$ |
| $N = 8$ | Number of snippet capsules |
| $M = 80$ | Number of segment capsules |
| **Snippet Capsule** | |
| SniCap | Snippet Capsule |
| $\mathcal{T}$ | Snippet template (of a snippet capsule) |
| $\boldsymbol{A}$ | Snippet transformation parameters |
| $\mu_i$ | Activation of the $i^{th}$ snippet template |
| **Segment Capsule** | |
| SegCap | Segment Capsule |
| $\mathcal{P}$ | Segment template (of a segment capsule) |
| $\boldsymbol{P}$ | Spatial relation between a segment template and all the snippet templates |
| $\boldsymbol{\alpha}$ | The weights of snippet templates when used to form a segment template |
| $\boldsymbol{B}$ | Segment transformation parameters |
| $\nu^{(k)}$ | Activation of the $k^{th}$ segment template |

## B  Number of Layers

The necessity of a two-layer hierarchy is briefly discussed in Section 3.3. In short, it is difficult for a single-layer hierarchy to capture long-time dependencies and variations. This section describes an empirical study where we compare MCAE with its single-layer correspondence. The single-layer model is an MCAE without the segment autoencoder and with an increased number of 80 snippet capsules. The snippet length $l$ is set to input length $L$, and the snippet activations $\mu$ are used for contrastive learning. The double-layer model is the MCAE proposed in the paper. Both models are

35th Conference on Neural Information Processing Systems (NeurIPS 2021).

Table A2: Performance of single and double layer model on T20 dataset of different clip lengths.

| | $L = 32$ | $L = 64$ | $L = 128$ | $L = 256$ |
|---|---|---|---|---|
| Single | $47.64 \pm 1.68$ | $28.43 \pm 2.95$ | $35.23 \pm 1.38$ | $31.42 \pm 0.53$ |
| Double | $69.30 \pm 0.76$ | $69.88 \pm 3.53$ | $66.45 \pm 0.39$ | $65.24 \pm 6.62$ |

Table A3: Convolution backbone. When used in the snippet encoder, $C = 8$ and $D = 5N$. When used for 1D-Conv baseline, $C = 48$ and $D$ is set to the number of hidden units. The Leaky ReLU has a negative slope of 0.01.

| | Channels in | Channels out | Kernel Size | Stride | Padding |
|---|---|---|---|---|---|
| 1D Conv Layer | 2 | $C$ | 4 | 2 | 1 |
| Batch Normalization | | | | | |
| Leaky ReLU | | | | | |
| 1D Conv Layer | $C$ | $2C$ | 4 | 2 | 1 |
| Batch Normalization | | | | | |
| Leaky ReLU | | | | | |
| 1D Conv Layer | $2C$ | $4C$ | 4 | 2 | 1 |
| Batch Normalization | | | | | |
| Leaky ReLU | | | | | |
| 1D Conv Layer | $4C$ | $D$ | 1 | 1 | 0 |

trained using samples from T20 interpolated to four different lengths $\{32, 64, 128, 256\}$. The results are shown in Table A2. The first observation is that the single-layer model performs poorly in all four configurations. More importantly, as $L$ increases, the single-layer model degrades severely while the double-layer model performs well consistently.

## C  Implementation

**Transformation Parameters**  The MCAE implementation in the main paper works in 2D spaces. To regulate the model, the snippet and segment encoders are set to output five parameters for each template: $\mu$ (or $\nu$), $s$, $t_x$, $t_y$, and $\theta$, where the first parameter is the activation and the last four parameters form a transformation as follows

$$\begin{pmatrix} \sigma(s)\cos\theta & -\sigma(s)\sin\theta & f(t_x, 1.5) \\ \sigma(s)\sin\theta & \sigma(s)\cos\theta & f(t_y, 1.5) \\ 0 & 0 & 1 \end{pmatrix}, \tag{1}$$

where $\sigma(\cdot)$ is the sigmoid function, and $f(x, t) = \max(-t, \min(x, t))$ "clamps" $x$ within $[-t, t]$. The value $t = 1.5$ allows for more flexibility as the input is generally in $[-1, 1]$ in our experiments.

**Snippet Encoder**  The 1D ConvNet $f_{\mathrm{conv}}$ in the snippet encoder is defined in Table A3, with $C = 8$ and $D = 5N$ where $N$ is the number of snippet capsules, and the factor 5 corresponds to the five parameters for each capsule $\{\mu, s, t_x, t_y, \theta\}$. The $5N$-dimension output is used as transformation parameters for snippet capsules.

**Segment Encoder**  The $f_{\mathrm{LSTM}}$ in the segment encoder is a bi-directional LSTM (BiLSTM) with 32 hidden units. The 64-dimension hidden state of $f_{\mathrm{LSTM}}$ at the last time step is sent to a fully connected layer which gives $32M$-dimension output. It is then fed into $M$ different fully connected layers, each of which outputs five parameters for a segment capsule.

**Baselines**  The 1D-Conv baseline uses the same architecture as $f_{\mathrm{conv}}$, which is defined in Table A3. To improve its performance, we use $C = 48$ and experimented with a variable $D$ as hidden unit numbers. The LSTM baseline is a BiLSTM with 256 hidden units. Its 512-dimension output is fed to a fully connected layer with a variable output dimension $D$, whose output is activated by a leaky ReLU function. Different values of $D$ have been explored in the main paper.

# D  Training Details

**Hyperparameters**  We set the batch size to 64 and use a fixed learning rate $10^{-3}$. The models are optimized using the Adam [1] optimizer. The training stops when the model's performance stagnates for over 100 epochs or right after the $1000^{th}$ epoch, whichever is earlier. For experiments on T20, we set the random seed to 0, 1, and 2. For experiments on NW-UCLA, NTURGBD60, and NTURGBD120, we set random seed 0. There is no tuning of random seeds. The loss weights $\lambda^{\texttt{Sni}}$ and $\lambda^{\texttt{Seg}}$ are searched in $\{0.5, 1, 1.5, 2, 5, 10\}$. For T20 and NW-UCLA, we use $\lambda^{\texttt{Sni}} = \lambda^{\texttt{Seg}} = 1$. For NTURGBD60 and NTURGBD120, we use $\lambda^{\texttt{Sni}} = 10$ and $\lambda^{\texttt{Seg}} = 5$.

**Software and Hardware**  All the models are implemented using PyTorch 1.8 compiled with CUDA 11.2 and CuDNN 7.6.5. The computation runs on an NVIDIA Titan V GPU with 12GB memory. The typical time required for experiments on T20, NW-UCLA, NTURGB60, and NTURGBD120 is 7hrs, 0.5hrs, 6hrs, and 16hrs, respectively.

**Contrastive Learning**  We use four disturbance (data augmentation) methods in the experiments:

1. **Rotate**: Applied to T20 and skeleton datasets. For T20 dataset, the input is rotated by a random angle between $-30°$ and $30°$. For skeleton datasets, the orientation for rotation is randomly determined from yaw, pitch, or roll.

2. **Smooth**: Applied to T20 and skeleton datasets. The input sequences are temporally filtered by a moving average kernel of size 3.

Table A4: Examples of the Trajectory20 dataset (a.t. is short for "asymptotic to").

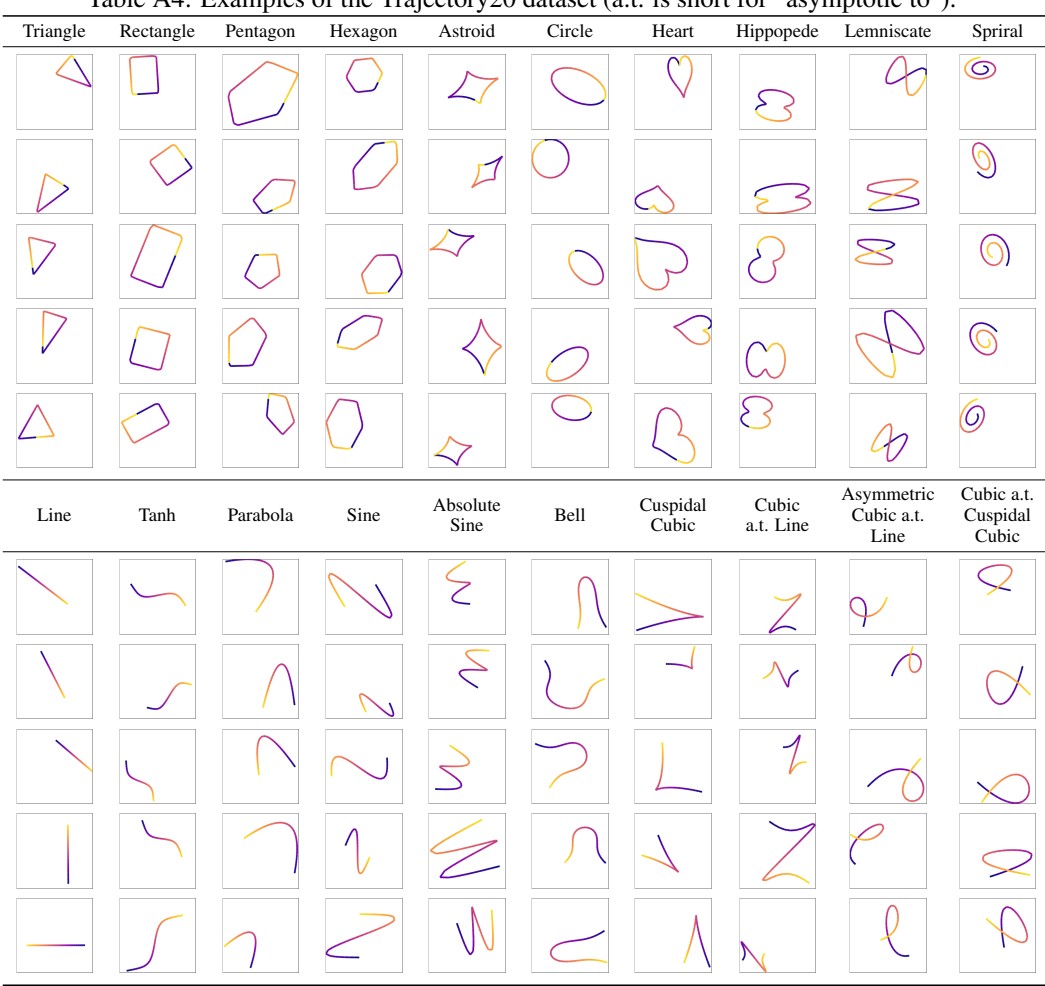

3. **Jittering**: Applied to skeleton datasets only. A random number $m \sim U(0, 1)$ is sampled for each joint. The coordinates of a joint are disturbed by Gaussian noise $n \sim \mathcal{N}(0, 1)$ in all the time steps if $m < 0.1$ and kept unchanged otherwise.

4. **Masking**: Applied to skeleton datasets only. A random number $m \sim U(0, 1)$ is sampled for each joint. The coordinates of a joint are masked by 0 in all the time steps if $m < 0.2$ and kept unchanged otherwise.

The disturbance methods applied for each training sample are randomly determined.

## E  Examples of the Trajectory20 (T20) dataset

We show in Table A4 some examples of the T20 dataset. The color gradient from blue to yellow indicates the time steps. For closed trajectories, the point moves in a randomly selected direction and finishes one whole trajectory. For open trajectories, the point starts at one end in a randomly selected direction and finishes at the other end.

## References

[1] Diederik P. Kingma and Jimmy Ba. Adam: A method for stochastic optimization. In *ICLR*, 2015.