# OpenReview forum: "Unsupervised Motion Representation Learning with Capsule Autoencoders"
_NeurIPS.cc/2021/Conference — NeurIPS 2021 Poster_

### Official Review · Reviewer_fGts · 2021-07-15

**Rating:** 5
**Confidence:** 4

**Summary:**

This paper presents an unsupervised motion representation learning method with capsule autoencoders. Its framework is composed of two-level motion modeling: snippet-level for short and local motion capturing, and segment-level for full-length semantic-aware motion capturing. It presents a new dataset called as Trajectory 20 and performs experiments on two datasets.

**Ethical Concerns:**

N/A.

**Limitations And Societal Impact:**

N/A.

**Main Review:**

1. The proposed autoencoder is a feed-forward network of temporal convolutions to predict the transformation parameter and activations. I cannot figure out the relation between the proposed autoencoder and the traditional capsule network. This paper needs to make a clear description on its work with respect to the capsule network.

2. This paper fails to present experiments on more common motion datasets, like movingMNIST, and thus I cannot clearly tell the advantages of the proposed method over previous ones.

3. In the ablation studies, the authors fail to provide a study on the importance of two-level motion modeling. For example, it should compare with a simple baseline of one-level modeling with the same number of snippets.

4. The evaluation datasets are quite clear and like toy examples. I would like to see more experiments on motion representation learning on more realistic datasets, like cloud points for scene flow estimation, RGB videos for optical flow estimation.

5. The idea of two-level motion modeling has appeared in many video action recognition works. For example: V4D:4D Convolutional Neural Networks for Video-level Representation Learning [ICLR 2020]. It should discuss these works in the related work section.


**Time Spent Reviewing:**

3

---

> ### Author Response · Authors · 2021-08-10
> **Thank you for the comments. Below are our responses about the relation between our work and capsule network, the use of datasets, experiment design, and discussion about prior works.**
>
> Thank you for the comments. Below are our responses about the relation between our work and capsule network, the use of datasets, experiment design, and discussion about prior works.
>
> ---
> > The proposed autoencoder is a feed-forward network of temporal convolutions to predict the transformation parameter and activations. This paper needs to make a clear description on its work with respect to the capsule network.
>
> To the best of our knowledge, so far there is no consensus on the best practice of generating and manipulating capsules. MCAE is feed-forward, which makes it different from the routing networks like [Sabour et al. 2017][Hinton et al. 2018] and more similar to models like [Kosiorek et al. 2019]. We will elaborate on this difference in the updated manuscript.
>
> We believe the idea and merit of capsules [Hinton et al. 2011] on image classification is that it separates the identity of an entity (a part, an object, or in our case a snippet/segment/motion) from its pose (or any other transformation/deformation), so that a part-whole structure can be learned and the entity effectively recognized. This has been explored by all the works above, and the proposed MCAE.
>
> ---
> > No experiments on more common motion datasets, like movingMNIST. Cannot clearly tell the advantages of the proposed method over previous ones.
>
> We have experiments on the commonly used skeleton action dataset. There are two reasons that made us propose the T20 dataset instead of using movingMNIST, as stated in L184-187. First, movingMNIST is innately linear and has limited motion variations. Second, it is used for prediction rather than classification.
>
> We show the efficacy of MCAE in the following two ways:
>
> (1) T20 dataset: although there’s no prior work on this proposed dataset, we compared MCAE with carefully designed LSTM/convolution-based baselines, and traditional time-series clustering methods.
>
> (2) Skeleton-based action dataset: we extended our model to MCAE-MP and compared it with prior works on the commonly used NTU60/120 and NW-UCLA dataset.
>
> ---
> > No ablation study on the importance of two-level motion modeling.
>
> In Section B of the supplementary material, we have an ablation study comparing double-level and single-level representation. We used the T20 dataset with different sequence lengths L. The results are as follows (copied from Table 2, supplementary material):
>
> |        |    L=32    |    L=64    |    L=128   |    L=256   |
> |--------|:----------:|:----------:|:----------:|:----------:|
> | Single | 47.64±1.68 | 28.43±2.95 | 35.23±1.38 | 31.42±0.53 |
> | Double | 69.30±0.76 | 69.88±3.53 | 66.45±0.39 | 65.24±6.62 |
>
> The first observation is that the single-layer model performs poorly in all four configurations. More importantly, as L increases, the single-layer model degrades severely while the double-layer model performs well consistently.
>
> ---
> > The evaluation datasets are quite clear and like toy examples, missing more realistic datasets like point clouds and RGB videos.
>
> We acknowledge the importance of research on more complex point systems (like point clouds) as a future extension of MCAE. However, we disagree that the datasets we used are clear and toy-like. We used both synthetic data (the proposed T20 dataset) and the skeleton-based action dataset (NTU60/120, NW-UCLA) in our experiment. The variation in T20 is more complex than movingMNIST and is non-trivial to model. The skeleton-based ones are widely used as challenging datasets. In our opinion, this covers both controlled environments (of a single point) and real-life human actions (of multiple moving joints).
>
> ---
> > No discussion about prior works that uses two-level motion modeling, e.g. [ICLR 2020].
>
> Thank you for the comment. We agree that two/multi-level motion modeling is widely used and plays an important role in long-term dependency modeling for videos. As shown in the ablation study in Section B (supplementary material), our model has also benefited from this design while learning transformation-invariant information at snippets/segments levels. We are committed to include discussion about V4D [Zhang et al. 2020], along with the related works like [Hussein et al. 2019][Wang et al. 2021], in the updated manuscript.
>
> ### References
> [Hussein et al. 2019] Noureldien Hussein, Efstratios Gavves, Arnold W. M. Smeulders: Timeception for Complex Action Recognition. CVPR 2019.
>
> [Hinton et al. 2011] Geoffrey E. Hinton, Alex Krizhevsky, Sida D. Wang: Transforming Auto-Encoders. ICANN 2011.
>
> [Hinton et al. 2018] Geoffrey E. Hinton, Sara Sabour, Nicholas Frosst: Matrix capsules with EM routing. ICLR 2018.
>
> [Kosiorek et al. 2019] Adam R. Kosiorek, Sara Sabour, Yee Whye Teh, Geoffrey E. Hinton: Stacked Capsule Autoencoders. NeurIPS 2019.
>
> [Sabour et al. 2017] Sara Sabour, Nicholas Frosst, Geoffrey E. Hinton: Dynamic Routing Between Capsules. NIPS 2017.
>
> [Wang et al. 2021] Limin Wang, Zhan Tong, Bin Ji, Gangshan Wu: TDN: Temporal Difference Networks for Efficient Action Recognition. CVPR 2021.
>
> [Zhang et al. 2020] Shiwen Zhang, Sheng Guo, Weilin Huang, Matthew R. Scott, Limin Wang: V4D: 4D Convolutional Neural Networks for Video-level Representation Learning. ICLR 2020.

---

### Official Review · Reviewer_B1sR · 2021-07-15

**Rating:** 6
**Confidence:** 3

**Summary:**

The paper presents a method for incorporating transformation invariance using capsule networks in the unsupervised learning of motion representations. The proposed method solves the problem in two steps, snippet and segment learning, which correspond to a lower and higher level of motion signals (basic trajectories -> reconstruction/time -> Contrastive learning and reconstruction). The method was evaluated on a synthetic dataset and three skeleton datasets for action recognition achieving convincing results.

**Main Review:**

Strengths:
-The paper is well-structured
-Experimental results are promising

Limitations:
- I found the description of the model very hard to follow (both the snipped and segment learning). For someone that is not familiar to CapsNet, it basically is impossible to understand what is happening. Basics of CapsNet should be recalled to build a parallel with the proposal.
-The authors noted a few works that employ capsule networks in motion representation in Related Works [3,30, 47] but did not attempt to compare against them in the experiments.
-2D projected inputs were used instead of 3D pose sequence, did the authors try to experiment with the original 3D sequences?
-It would be interesting to see what has the model learnt on the skeleton dataset.

Originality:
- The work is original. Capsule Nets have been designed especially to do this. However, few works that were cited introduced ways of representing "images". A comparison should have been made to see if the introduction of "motion" really is bringing something to the Table.

Clarity:
- The model Section should be clarified, making references to what a CapsNet is (formal definition).

Significance:
-The proposed method could have an impact on motion representation.

**Time Spent Reviewing:**

3

---

> ### Author Response · Authors · 2021-08-10
> **Thank you for the comments. Below are our responses to the raised issues regarding clarity/background of the CapsNet, comparison with prior works, and experiments.**
>
> Thank you for the comments. As a summary, below are our responses to the raised issues regarding clarity/background of the CapsNet, comparison with prior works, and experiments.
>
> ---
> > Basics of CapsNet should be recalled, the model section should be clarified with formal definition to a CapsNet.
>
> Thank you for the comment. We will improve the model section by adding the necessary preliminaries. Briefly, the idea of capsules [Hinton et al. 2011] on image classification is to separate the identity of an entity (a part, an object, or in our case a snippet/segment/motion) from its pose (or any other transformation/deformation), so that a part-whole structure can be learned and the entity is effectively recognized. In the proposed MCAE, the parameters for snippet and segment capsules have the same structure: a template (the identity), a transformation matrix (the “pose”), and an activation scalar. The capsules can be computed via either routing network [Sabour et al. 2017] or feed-forward networks [Kosiorek et al. 2019], and MCAE belongs to the latter family.
>
> ---
> > What does “motion” bring, compared with the capsule models proposed for images?
>
> On the one hand, literature about skeleton-based action recognition (i.e., [Cheng et al. 2020], [Liu et al. 2020], [Cai et al. 2021], and methods listed in Table 3) have evidenced the efficacy and efficiency of using pure motion signals without RGB frames in human action analysis. On the other hand, despite the intriguing transformation-invariance property of capsule networks, its application has been mainly confined within images. We believe that introducing motion to the capsule network, or the other way round, provides
>
> (1) A new, robust, and efficient view into motion signals in any dimension space under the transformation-invariance assumption (though the motion and transformation in $d>3$ dimension space could have semantics different from their 2/3D counterparts).
>
> (2) Proof that disentangling identity from transformation variance works not only for visual problems but a possibly larger family of time series analysis problems.
>
> ---
> > Works on CapsNet for motion representation [3,30,47] cited but not experimentally compared.
>
> Below is a conceptual comparison between MCAE and [3,30,47].
>
> |      | Supervised? |    Input Modality   | Network Structure |  Task |
> |:----:|:-----------:|:-------------------:|:-----------------:|:-----------------------:|
> |  [3] |     Yes     |      RGB Frames     |  Dynamic Routing  |       Recognition       |
> | [30] |     Yes     |      RGB Frames     |  Dynamic Routing  | Video Error Concealment |
> | [47] |     Yes     |       Skeleton      |  Dynamic Routing  |       Recognition       |
> | MCAE |      No     | Trajectory/Skeleton |    Autoencoder    |       Recognition       |
>
> We are unable to experimentally compare them with MCAE because:
>
> (1) MCAE is an unsupervised model (autoencoder), whereas [3,30,47] are supervised models.
>
> (2) The motion represented by frame transition (in [3,30]) and coordinate changes (in MCAE) are semantically different due to different input modality. Therefore, a quantitative comparison could be unfair.
>
> (3) Although [47] reported results on NTU-RGBD60, it is (a) supervised, and (b) obtained on a self-curated and very limited subset, which is not made public.
>
> We will provide a more detailed conceptual comparison for clarification.
>
> ---
> > Did the authors experiment with the original 3D sequences?
>
> Yes, we did. We found that MCAE learned with the original 3D sequences performed worse than using 2D projected sequences. In our preliminary experiment, MCAE learned with 3D sequence using linear auxiliary classifiers (SLP) gives 51.9 and 54.2 on NTU60-XSUB and NTU120-XSET, respectively.
>
> We have the following conjectures regarding why 2D models outperform 3D ones:
>
> (1) Our snippet/segment transformations are implemented as similarity transformations. Three 2D similarity transformations directly allow non-uniform scale change, whereas one 3D similarity transformation does not.
>
> (2) Using three 2D transformations decouples transformation parameters, while one 3D transformation multiplies them together. The multiplication prohibits direct gradient feedback to the parameters, making learning more difficult. Below shows an example of the rotation matrix in 3D, where parameters $\alpha$, $\beta$, and $\gamma$ are coupled.
>
> \\begin{align*}
> R &= R_{z}(\\alpha) R_{y}(\\beta) R_{x}(\\gamma) \\\\
> & =
> \\begin{pmatrix} cos\\alpha & - sin\\alpha & 0 \\\\ sin\\alpha & cos\\alpha &0 \\\\ 0 & 0 & 1 \\end{pmatrix}
> \\begin{pmatrix} cos\\beta & 0 & sin\\beta \\\\ 0 & 1 & 0 \\\\ -sin\\beta & 0 & cos\\beta \\end{pmatrix}
> \\begin{pmatrix} 1 & 0 & 0 \\\\ 0 & cos\\gamma & -sin\\gamma \\\\ 0 & sin\\gamma & cos\\gamma \\end{pmatrix} \\\\
> & =
> \\begin{pmatrix}
> cos\\alpha cos\\beta & cos\\alpha sin\\beta sin\\gamma - sin\\alpha cos\\gamma & cos\\alpha sin\\beta cos\\gamma + sin\\alpha sin\\gamma \\\\
> sin\\alpha cos\\beta & sin\\alpha sin\\beta sin\\gamma + cos\\alpha cos\\gamma & sin\\alpha sin\\beta cos\\gamma - cos\\alpha sin\\gamma \\\\
> -s \\beta & cos\\beta sin\\gamma & cos\\beta cos\\gamma
> \\end{pmatrix}
> \\end{align*}
>
> ---
> > What does the model learn on the skeleton dataset?
>
> The snippets are similar to what have been learned on T20. The segments are less interesting than their counterparts in T20. This is because:
>
> (1) A skeleton action class is marked by very few joints and most of the joints stay still during an action.
>
> (2) The motion of the joints are modeled separately and their segment-level representations are concatenated for final classification. This makes the segments not directly related to specific motion classes, unlike the T20 case.
>
> We agree that examples and discussions about the segments learned on skeleton are important, and we plan to add it to the updated supplementary materials.
>
> ### References
> [Cai et al.] Jinmiao Cai, Nianjuan Jiang, Xiaoguang Han, Kui Jia, Jiangbo Lu:
> JOLO-GCN: Mining Joint-Centered Light-Weight Information for Skeleton-Based Action Recognition. WACV 2021.
>
> [Cheng et al. 2020] Ke Cheng, Yifan Zhang, Congqi Cao, Lei Shi, Jian Cheng, Hanqing Lu:
> Decoupling GCN with DropGraph Module for Skeleton-Based Action Recognition. ECCV 2020.
>
> [Hinton et al. 2011] Geoffrey E. Hinton, Alex Krizhevsky, Sida D. Wang: Transforming Auto-Encoders. ICANN 2011.
>
> [Hinton et al. 2018] Geoffrey E. Hinton, Sara Sabour, Nicholas Frosst: Matrix capsules with EM routing. ICLR 2018.
>
> [Kosiorek et al. 2019] Adam R. Kosiorek, Sara Sabour, Yee Whye Teh, Geoffrey E. Hinton: Stacked Capsule Autoencoders. NeurIPS 2019.
>
> [Liu et al. 2020] Ziyu Liu, Hongwen Zhang, Zhenghao Chen, Zhiyong Wang, Wanli Ouyang:
> Disentangling and Unifying Graph Convolutions for Skeleton-Based Action Recognition. CVPR 2020.
>
> [Sabour et al. 2017] Sara Sabour, Nicholas Frosst, Geoffrey E. Hinton: Dynamic Routing Between Capsules. NIPS 2017.

---

### Official Review · Reviewer_Vi9D · 2021-07-16

**Rating:** 7
**Confidence:** 3

**Summary:**

This work proposes a motion representation learning method based on capsule autoencoders. It decomposes an input 2D keypoint trajectory into snippets and segments, and form a hierarchical representation that is also transformation invariant. The proposed motion representation achieves state-of-the-art motion classification results on a self-proposed synthetic 2D motion dataset as well as skeleton-based human action recognition.

**Ethical Concerns:**

The authors have adequately acknowledged the ethical concerns of their work.

**Limitations And Societal Impact:**

Yes, the authors have adequately addressed the limitations and potential negative societal impact.

**Main Review:**

Strength:

- The proposed method is based on the important insight that movement patterns should be transformation invariant and also should naturally consist of a hierarchy of patterns. Thus, the proposed method is intuitive and easy to understand. To the best of my knowledge, using the formulation of capsule autoencoders to explicitly learn transformable trajectory templates through self-supervised learning is novel and elegant. Such formulation can also provide an interpretable representation for future motion analysis tasks.
- The paper conducts extensive experiments and studies that backed up its claims:
    - Experiments on the synthetic T20 dataset suggest that the proposed method is an effectively self-supervised motion representation learning method.
    - Real-world applications on self-supervised action recognition also suggest the strength of the proposed method.
    - Ablation studies that specify the interpretability of the proposed method: I find section 4.3 especially helpful in showcasing the benefit of the proposed method, where the transformation invariance is well-documented.
- The proposed T20 dataset would be beneficial as a standard test ground for future 2D motion analysis methods.

Issues:

- In section 4.3, it seems old that theta is constrained to [-10, 10] degrees, where a hexagon should be expected to activate similar segments for large rotation perturbation?
- To what extent can a motion trajectory be transformed but still be classified as a class? The proposed method seems to be confined to 2D similarity transformation instead of the full affine/projective transformation.
- (Minor) It would be beneficial to include state-of-the-art results on supervised skeleton-based action classification, to give readers an idea of how far unsupervised methods are away from the supervised ones.
- (Minor) It would be beneficial if additional ablation on the effect of the contrastive learning loss could be provided. While conceptually the contrastive learning loss would be beneficial in learning the segments, I am curious to see its effects compared to only using reconstruction loss.

====================================================================================================

After the author response:

The answers from the authors are largely satisfactory to me (concerning robustness with rotation, contrastive loss). I think the proposed MCAE would be a worthy contribution to the current capsule network literature. I would like to keep my original score.

**Time Spent Reviewing:**

5

---

> ### Author Response · Authors · 2021-08-10
> **Thank you for the comments. Below are our responses to the issues about transformation robustness and experiments.**
>
> Thank you for the comments. Below are our responses to the issues about transformation robustness and experiments.
>
> ---
> > Will a hexagon activate similar segments for large rotation perturbation?
>
> Yes, we observe that the activated segments are generally consistent (despite slight differences) for all motion classes when the rotation is 30 degrees. For even larger rotations, we usually observe changes. This is because we allow multiple segments to be activated for an input and the segments are overcomplete for the classes in T20. In the paper $\theta$ is confined within [-10, 10] to guarantee the consistency of segment activation so that the rotation captured by segments can be well illustrated. We will extend this study to larger rotations and provide analysis in the updated supplementary materials.
>
> ---
> > To what extent can a motion trajectory be transformed but still be classified as a class?
>
> This is an interesting question. The transformation parameters of snippets/segments are 2D similarity transformations. However, MCAE is able to handle affine and possibly more general nonlinear transformations (we do not have enough empirical results for the latter yet). This is because
>
> (1) we snippet-wise apply the transformation parameters and the aggregated effect on segments is in general beyond a similarity transformation, and
>
> (2) we allow multiple activated segments (whose weights are close to 1) for an input. The results on the T20 dataset (generated using affine transformation) provides evidence for the statements above.
>
> ---
> > State-of-the-art results of supervised results for skeleton-based action recognition.
>
> Thank you for the comment. We plan to add the supervised results from recent publications, as follows:
>
> |                     | NTU60-XSUB | NTU60-XVIEW | NTU120-XSUB | NTU120-XSET | NW-UCLA |
> |---------------------|:----------:|:-----------:|:-----------:|:-----------:|:-------:|
> | [Cheng et al. 2020] |    90.8    |     96.6    |     82.4    |     84.3    |   93.8  |
> | [Liu et al. 2020]   |    91.5    |     96.2    |     86.9    |     88.4    |    -    |
> | [Cai et al. 2021]   |    93.8    |     98.1    |     87.6    |     89.7    |    -    |
>
> ---
> > Ablation study on the effect of contrastive learning loss?
>
> Following your comment, we trained MCAE without contrastive loss and observed severe performance drop. We set $L_{con}^{Seg}$ to 0 and $L_{sps}^{Reg}$ (the sparsity regularizer) to {0.1, 0.2, 0.5, 1.0}. The results are as follows.
>
> | Sps. Reg. |     0.1    |     0.2    |     0.5    |     1.0    |
> |:---------:|:----------:|:----------:|:----------:|:----------:|
> |    Acc.   | 13.36±0.72 | 12.73±0.22 | 15.08±2.02 | 17.07±1.38 |
>
> While increasing sparsity regulation improves performance, it (along with reconstruction loss) provides little feedback about the spatial transformation. Contrastive loss plays a vital role in the learning process by providing such information. However, proper model design that captures the spatial information is equally important. This can be seen in Table 2 in both main text and supplementary material, where the baselines are trained with the same contrastive loss but have inferior performance. We will add this discussion in the updated manuscript.
>
> ### References
>
> [Cai et al. 2021] Jinmiao Cai, Nianjuan Jiang, Xiaoguang Han, Kui Jia, Jiangbo Lu:
> JOLO-GCN: Mining Joint-Centered Light-Weight Information for Skeleton-Based Action Recognition. WACV 2021.
>
> [Cheng et al. 2020] Ke Cheng, Yifan Zhang, Congqi Cao, Lei Shi, Jian Cheng, Hanqing Lu: Decoupling GCN with DropGraph Module for Skeleton-Based Action Recognition. ECCV 2020.
>
> [Liu et al. 2020] Ziyu Liu, Hongwen Zhang, Zhenghao Chen, Zhiyong Wang, Wanli Ouyang:
> Disentangling and Unifying Graph Convolutions for Skeleton-Based Action Recognition. CVPR 2020.

---

### Official Review · Reviewer_8Ljw · 2021-07-16

**Rating:** 7
**Confidence:** 4

**Summary:**

This work presents a Motion Capsule Autoencoder (MCAE) which learns to represent motions. The two-staged architecture first learns a set of snippet capsules, which learn motions for short time-spans, and then learns segment capsules that learn the long-term motions from all the snippet capsules. These capsules are learned end-to-end in an unsupervised manner with reconstruction losses and a contrastive loss. The approach is evaluated on a synthetic motion dataset proposed in this work, Trajectory20, as well as on three real-world skeleton-based human action recognition datasets, where it is shown to achieve strong performance when compared to previous state-of-the-art methods.

**Limitations And Societal Impact:**

The authors address potential negative societal impacts, but could include more discussion about the limitations. For example, what type of motion patterns does the MCAE not perform well on? Can the method generalize to longer sequences (i.e. $L>32$)?

**Main Review:**

Overall, this is an interesting work which presents a novel capsule architecture (MCAE) to learn motion patterns in an unsupervised manner. Their experimental evaluations on both synthetic and real data, comparisons with previous approaches, and analysis all support the strengths of this work. I suggest that is paper should be accepted, but the following questions/comments should be addressed to improve the clarity of the work.

1) How are the templates learned/obtained? They are defined as $ \mathcal{T} = t_i | t_i \in \mathbb{R}^d, i=1,..., l $. Are they a set of learned parameters, which are trained with the full system in an end-to-end manner? Also, $d$ is not defined - since the network works on 2d points, and $A$ is a $3\times3$ matrix, I assume $d=2$, but it should be stated explicitly.

2) In the ablation study, how is the accuracy obtained? Later in lines 227-228, it states that an MLP is learned on top of the model for the motion classification experiments. Is it the same for the ablations? If so, this information should be included in the, or prior to the ablations section.

3) How is the LSTM used in Eq. 3 to encode the SniCaps? Since $C_i^\text{Sni}$ is a set of tuples $(A_i,\mu_i)$, what operation is used to combine each $C_i^\text{Sni}$ into a single representation/vector?

4) In Section 4.3, the authors state that Table 4 shows that the top-5 segment templates for the hexagon remain consistent across all listed rotations, but changes for abs_sine (e.g. IDs 33 and 32 are present in $\theta=-10$ and $\theta=10$ respectively). Is this due to the nature of the motion pattern (i.e. the abs_sine's open trajectory rather than the hexagon's closed trajectory), or is such behavior common for many different motion classes?

5) Is $\mu$ the result of a sigmoid or softmax activation?

**Time Spent Reviewing:**

5

---

> ### Author Response · Authors · 2021-08-10
> **Thank you for the comments. Below are our responses to the raised issues about model implementation and experiments.**
>
> Thank you for the comments. Below are our responses to the raised issues about model implementation and experiments. We will update the manuscript accordingly for better clarification.
>
> ---
> > How are the templates obtained?
>
> The templates are learned end-to-end with the full system.
>
> ---
> > $d$ is not defined.
>
> We agree that $d$ should have been made clear when introduced. We used $d=2$ in all our experiments. Notably, $d$ can be set to other values if the motion is in higher-dimensional spaces.
>
> ---
> > How is the accuracy obtained in ablation study?
>
> It is the same as in lines 227-228. Following your comment, we will rearrange the sentences in L227-228 to L179, where we state “All unsupervised accuracies are produced by an auxiliary linear classifier that is trained on the motion representation learned by MCAE or the baselines, but whose gradient is blocked from back-propagating to the model.”
>
> ---
> > How is the LSTM used in Eq. 3 to encode SniCaps?
>
> We convert $(A_i, \mu_i)$ into a single vector by (1) flatten transformation $A_i$ into a 6-dimensional vector $A_i’$ (the last row of $A_i$ (a 3x3 matrix) are constants and thus discarded), and (2) concatenate $A_i’$ with $\mu_i$. This results in a (6+1)N-dimensional vector for a SniCap. We will update the manuscript to clarify this.
>
> ---
> > Regarding Table 4: is the behavior (changes in top-5 segments) common for different motion classes?
>
> This phenomenon is observed in many different motion classes, both open and closed. Our interpretation to this is that the segments are overcomplete for T20 classes, and allows for slightly different segments activated when the sample is transformed.
>
> ---
> > Are $\mu$ and $\nu$ obtained by sigmoid or softmax?
>
> We choose to use sigmoid to obtain $\mu$ by empirical study. Intuitively, sigmoid activates multiple snippets/segments and enables our model to better handle complicated motion signals.
>
> ---
> > Could include more discussion about the limitations, e.g. (1) What type of motion patterns does the MCAE not perform well on? (2) Can the method generalize to longer sequences?
>
> Thank you for the comments. We plan to provide further analysis on:
>
> (1) Failure cases: below are class-level accuracies.
>
> The accuracy for closed patterns:
>
> | | | | | |
> |:--------:|:---------:|:--------:|:---------:|:--------:|
> | triangle | rectangle |  pentagon  | hexagon | astroid  |
> |   73.6   |    36.8   |    22.2    |   33.8  |   56.2   |
> |  circle  | hippopede | lemniscate |  heart  | __average__|
> |   23.6   |    25.8   |    92.2    |   27.6  |   43.5   |
>
> The accuracy for open patterns:
>
> | | | | | | |
> |:------:|:-----------:|:-------------:|:-----------:|:---------------:|:--------:|
> | spiral |     sine    |    abs_sine   |     tanh    |       line      | parabola |
> |  99.8  |     99.2    |      99.2     |     89.8    |       97.6      |   75.8   |
> |  bell  | cusp. cubic | cubic at line | asym. cubic | cubic at cuspi. | __average__|
> |  78.6  |     86.4    |      99.6     |     99.2    |       98.2      |   93.0   |
>
> We found two confusion clusters in the closed patterns: (a) the *circle* cluster: *circle*, *rectangle*, *pentagon*, *hexagon*, and *hippopede*, and (b) the *triangle* cluster, which consists of *triangle* and *heart*. We further found that the sharp turns are smoothed when the templates are mixed for reconstruction, which could be the cause for the confusion above.
>
> (2) Sequences of different lengths: we evaluated MCAE on the T20 dataset with different lengths (32, 64, 128, and 256) in the comparison between single-layer and double-layer models (Section B, supplementary material). The results there show that MCAE performs reasonably well for longer sequences from T20. Please see the copied table below:
>
> | Model  |    L=32    |    L=64    |    L=128   |    L=256   |
> |--------|:----------:|:----------:|:----------:|:----------:|
> | Single | 47.64±1.68 | 28.43±2.95 | 35.23±1.38 | 31.42±0.53 |
> | Double | 69.30±0.76 | 69.88±3.53 | 66.45±0.39 | 65.24±6.62 |
>
> There are two observations:
> (a) The single-layer model performs poorly in all four configurations.
> (b) As $L$ increases, the single-layer model degrades severely while the double-layer counterpart performs well consistently.

---

### Decision · Program_Chairs · 2021-09-27

**Decision:**

Accept (Poster)

**Comment:**

This work introduces a novel capsule architecture that learns motion patterns in an unsupervised manner. Evaluation on synthetic and real-work skeleton-based human action datasets demonstrate the model good performance when compared to previous state-of-art methods.

Using the formulation of capsule autoencoders to explicitly learn transformable trajectory templates through self-supervised learning is novel and an interesting addition to the capsule network literature.